# The Effect of Extensive Human Presence at an Early Age on Stress Responses and Reactivity of Juvenile Ostriches towards Humans

**DOI:** 10.3390/ani8100175

**Published:** 2018-10-05

**Authors:** Pfunzo T. Muvhali, Maud Bonato, Anel Engelbrecht, Irek A. Malecki, Denise Hough, Jane E. Robinson, Neil P. Evans, Schalk W. P. Cloete

**Affiliations:** 1Department of Animal Sciences, University of Stellenbosch, Private Bag X1, Matieland 7602, South Africa; 20501609@sun.ac.za (P.T.M.); imalecki@iinet.net.au (I.A.M.); schalkc@elsenburg.com (S.W.P.C.); 2Directorate Animal Sciences, Western Cape Department of Agriculture: Oudtshoorn Research Farm, P.O. Box 351, Oudtshoorn 6620, South Africa; anele@elsenburg.com; 3School of Agriculture and Environment, Faculty of Science, The University of Western Australia, 35 Stirling Highway, Crawley, WA 6009, Australia; 4Institute of Biodiversity Animal Health and Comparative Medicine, University of Glasgow, Glasgow G12 8QQ, UK; Denise.Hough@glasgow.ac.uk (D.H.); Jane.Robinson@glasgow.ac.uk (J.E.R.); Neil.Evans@glasgow.ac.uk (N.P.E.); 5Directorate Animal Sciences, Western Cape Department of Agriculture: Elsenburg, Private Bag X1, Elsenburg 7607, South Africa

**Keywords:** welfare, stress responses, behaviour, ostrich, *Struthio camelus*

## Abstract

**Simple Summary:**

Husbandry practices for rearing ostriches in commercial farming environments are currently not optimized. Ostrich chicks may experience stressful episodes and fear of humans during routine farm management practices such as handling, which ultimately may impact on their welfare and explain the poor production performance observed in this species. However, extensive human presence and regular gentle handling has been demonstrated to alleviate stress sensitivity during handling by lowering the fear of humans in other species; be they kept as livestock, in a laboratory, or as pet animals. In this study, ostrich chicks exposed to extensive human presence and gentle handling showed lower stress sensitivity when handled for feather harvesting and clipping and were more inclined to associate with familiar humans at a later stage of their life compared with chicks that had limited human presence and care. This suggests that providing ostrich chicks with extensive human presence and gentle handling at a young age can assist in improving ostrich welfare.

**Abstract:**

The effect of extensive human presence and regular gentle handling performed at an early age (0–3 months old) on stress responses and reactivity of juvenile ostriches towards humans was investigated. A total of 416 ostrich chicks over two years were exposed to one of three treatments for three months after hatching; namely, Human Presence 1 (HP1, *N* = 144): extensive/prolonged human presence with physical contact (touch, stroking), gentle human voice, and visual stimuli; Human Presence 2 (HP2, *N* = 136): extensive/prolonged human presence without physical contact, but with gentle human voice and visual stimuli; and the Standard treatment (S, *N* = 136): human presence limited to routine feed and water supply as a control. At 7.5 months of age, the plasma heterophil/lymphocyte (H/L) ratio was measured before and 72 h after feather harvesting and feather clipping to determine acute stress responses, while chronic stress was measured by quantification of corticosterone (CORT) concentrations in the floss feathers of the birds. Birds’ behavioural response towards a familiar or an unfamiliar handler was evaluated at 12 months using docility and fear tests, and through behavioural observations conducted on random days between the ages of 8–13 months. Willingness to approach, and to allow touch interactions, aggressiveness, and exhibition of sexual display towards the handler, was recorded. No difference in the H/L ratios before and after feather harvesting and clipping was observed in HP1 birds, whereas H/L ratios showed a significant increase 72 h post feather harvesting and clipping in HP2 and S birds (*p* < 0.05). Birds from the S treatment exhibited a significantly (*p* < 0.05) higher feather CORT concentration compared with HP1 birds, while HP2 birds had intermediate responses. Birds’ reactivity towards humans and temperament as evaluated using behavioural observations, docility, and fear tests was not affected by treatment (*p* > 0.05). However, HP1 and HP2 birds were more inclined (*p* < 0.05) to approach a familiar rather than an unfamiliar handler during the behavioural observations, indicating an ability to distinguish between a familiar and an unfamiliar handler. Overall, the results indicate that early gentle human interactions with ostrich chicks can be beneficial in reducing physiological stress sensitivity later in life and facilitate the ability of ostriches to distinguish between familiar and unfamiliar handlers.

## 1. Introduction

Despite an increase in research efforts [1,2], the management of ostriches on commercial farms remains a challenge. This is partly because husbandry practices have not been optimised [3] and replacement breeders are not selected based on stress resistance or docile behaviour [4]. Consequently, the behaviour exhibited by mature birds often compromises not only the welfare of the birds, but also the health and safety of their handlers [2,5].

Positive human–animal interactions, when incorporated into normal husbandry practices, have been demonstrated to reduce physiological stress sensitivity and animals’ fear of humans, while improving docility and production in a range of livestock species including cattle [6] and pigs [7,8,9]. When exposed to stressful and fearful conditions, animals employ an adaptive mechanism to cope with the stressor, such as escaping the potential stressor. However, if the animal cannot escape and/or the stressor persists, physiological responses to stress are initiated. This includes the activation of the hypothalamic–pituitary–adrenal (HPA) axis, which regulates the circulating concentrations of corticosteroid hormones [10]. Previous studies have revealed that positive human–animal interactions were associated with lowering the circulating concentrations of corticosteroids [11,12]. For instance, in broiler chickens, early gentle handling lowers the heterophil/lymphocyte ratio and plasma corticosterone concentrations when exposed to a known stressor such as transportation [13]. It is thus possible that positive human handling may alter activity within, or the responsiveness of, the HPA axis in ways that may be beneficial to welfare across a range of domestic animal species [14]. The positive effects of human–animal interactions were also associated with improved docility, a lack of fear of humans, and an inclination to associate with humans, suggesting that such animals had a calmer temperament. Interestingly, positively handled dairy heifers kept a greater distance when approached by an unfamiliar handler [12]. This result suggests that they could distinguish between handlers and modify their behaviour accordingly. In commercial poultry, regular gentle handling of broiler chicks reduced their withdrawal response from an approaching human at a later stage of their life [15].

Preliminary work by Bonato et al. [16] demonstrated that ostrich chicks exposed to husbandry treatments that incorporated extensive human–chick interactions were more inclined to associate with humans at a later age than those with limited or no human presence. The aim of this study was thus to investigate the effect of three different husbandry treatments with varying levels of human presence and interactions performed at an early age on physiological stress responses and reactivity of ostriches towards humans during routine management practices when birds attained the juvenile stage and at puberty.

## 2. Materials and Methods

### 2.1. Study Population

The study was conducted at the Oudtshoorn Research Farm of the Western Cape Department of Agriculture, South Africa (33°63′ S, 22°25′ E). The study population, as well as the management of the breeding flock from which the experimental chicks originated, has been described elsewhere [1,17]. Briefly, the resource flock consisted of breeding pairs (*N* = 150 pairs) of three purebred ostrich breeds, South African Blacks (SAB), Zimbabwean Blues (ZB), Kenyan Reds (KR), and combinations of SAB with ZB and KR, respectively, as well as their reciprocal crosses. Eggs were collected daily, identified according to camp of origin, stored, and set weekly for artificial incubation. Each batch of chicks that hatched consequently consisted of a mix of breeds. Two batches of chicks were used to repeat the experiment over two years (2013 and 2015). After hatching, the chicks were tagged and identified with small numbered ostrich chick tags in the neck that were replaced with new numbered tags when the chicks reached five months of age. This was done to ensure that chick identity was unknown to the person conducting the successive reactivity tests to avoid bias. Feed and water were supplied *ad libitum* throughout the experiment in all treatments. Ethical clearance to conduct this study was granted by the Western Cape Department of Agriculture’s Departmental Ethical Committee for Research on Animals (Ref No.: R13/81).

### 2.2. Treatments

Two batches of chicks hatched in 2013 and 2015 were randomly allocated to one of three treatments, which were performed from hatching until three months of age. Treatments were as follows: Human Presence 1 (HP1)—chicks were exposed to extensive/prolonged human presence, gentle human voice, and visual contact along with gentle physical contact (touching, stroking on the body) (*N* = 68 and 76 for 2013 and 2015, respectively). Human presence 2 (HP2)—chicks were exposed to extensive/prolonged human presence with gentle human voice and visual contact, but without physical contact (*N* = 66 and 70 for 2013 and 2015, respectively). The Standard husbandry practice (S) acted as control, where human presence, human voice, and visual contact were limited to the supply of feed and fresh water when needed, without physical contact [18] (*N* = 66 and 70 for 2013 and 2015, respectively). Because ostriches are inquisitive animals, they tend to approach and explore humans frequently. This behaviour was thus used as an advantage in this study and chicks that approached the human handler in the HP1 group were gently touched and stroked as they came closer to the human who was sitting or walking in the enclosure. No chicks were forcefully touched or stroked as it could have induced stress. The HP1 and HP2 treatment were performed by four people (three adult females and one adult male) who interchanged with each other daily over the two years. All handlers were wearing similar green overalls to reduce potential discrimination from the chicks. Chicks were kept in an intensive chick rearing facility in enclosures within pens of 27.75 m^2^ (7.5 m × 3.7 m). Pens are divided by 0.5-m high walls, with three pens per room. HP1 and HP2 were kept in the same room, separated by one empty pen, while the S treatment was kept in a separate room to prevent visual contact with the other treatments. Chicks were kept indoors for the first week after hatching; thereafter, chicks were also allowed access to the camps outside (dry runs) of approximately the same size when the weather was permitting. The duration of human exposure for the HP1 and HP2 treatments was 100% of daylight hours (06:00–18:00, 12 h) during the first week after hatching (week 0: day 0–6). From week 1 (day 7–13), chicks were visited for 1 h at a time, with intervals increasing weekly by 1 h; therefore, during week 2 (day 14–20), chicks were visited for 1 h every 2 h, week 3 (day 21–27) chicks were visited for 1 h every 3 h, and so on (see Figure A1). From week 8 of the experiment, chicks were only visited for 1 h early in the morning (07:00 to 08:00) and during the afternoon (16:00–17:00), until week 12 when treatments were terminated. Thus, in total, HP1 and HP2 chicks received 343 h of human presence in the three-month duration of the experiment. Assuming that the supply of feed and water in the S treatment group was 30 min/day, S chicks received approximately 48 h of human visual contact during the three-month period. At three months of age, all groups were mixed together, and thereafter were reared as one group according to the standard husbandry practice of the farm. 

### 2.3. Physiological Measures of Stress

Acute (heterophil/lymphocyte: peak H/L ratio) and chronic (peak feather corticosterone) measures of physiological stress were assessed at 7.5 months of age as birds underwent feather harvesting (2013 and 2015 groups; *N* = 238) and feather clipping (2015 group only; *N* = 87). Feather harvesting and clipping are typically performed at this stage of their life as it corresponds to the transition from chick feathers to juvenile feathers when feathers are fully mature and ready for removal [19]. During feather harvesting, birds were caught and restrained, and feathers gently pulled by experienced stockmen not involved in the treatment phase [19]. For feather clipping, birds were also caught and restrained and feathers were clipped by the same stockmen above the bloodline (approximately 2.5 cm from their base), using pruning shears [20]. Handlers interacting with the birds during the first three months of their life were not involved in any handling of the birds to avoid association with potential stressful handling, which might affect successive behavioural tests. A drop of blood was collected from the wing vein before and 72 h after feather removal (*N* = 325) for assessment of the basal and peak stress responses, by means of the effects on the H/L ratio [21]. The H/L ratio has been shown to be a good indicator of acute handling stress in ostriches and can reflect the stress responses for up to four days (96 h) after handling [21]. Hence, on the day of collection, a blood smear was generated, air-dried, fixed by immersion in 99.9% methanol for 3 min, stained with 10% Giemsa solution for 45 min, rinsed, and air dried before being stored for later analysis. The H/L ratio was then assessed by counting one hundred white blood cells per slide by an investigator not involved in the experiment to avoid bias.

To evaluate activity within the HPA axis, as a result of routine farm management practices (i.e., weighing, tagging, health inspection, general human presence, movement/translocation of chicks) and environmental changes, corticosterone (CORT) concentrations were quantified in harvested floss feathers from 48 SAB ostriches (2015 group only; 16 from each treatment; 8 males and 8 females, respectively) using an adaptation of the methods of Bortolotti et al. [22]. Briefly, after removal of the calamus, the feathers were measured and cut into three equal sections (bottom, middle, and top). Each section was weighed, and the rachis removed. Only the top (growth during the treatment period) and bottom sections were used for analyses. To remove any external contaminants, the sections were washed with water and ethanol for 20 and 10 s, respectively. Samples were then cut into fine pieces (less than 5 mm) and allowed to air dry, before 20 mg was placed in labelled 20 mL borosilicate vials. A volume of 10 mL methanol was added to each vial before they were sonicated for 20 min and placed in a shaking incubator overnight for 17 h at 52 °C at 100 rotations per minute. From each vial, 8 mL of methanol was retained prior to the samples being washed twice with 2.5 mL of methanol, 2 mL of the wash was recovered each time and added to the original 8 mL of recovered methanol. The pooled extracts (12 mL) were filtered through 10 mL plastic syringes attached to 0.45 μm Ministart^®^ high flow syringe filters (Sartorius AG, Göttingen, Germany) and collected in 12.5 mL borosilicate test tubes. Samples were evaporated to dryness using a sample concentrator at 48 °C in a standard airflow fume hood. Samples were reconstituted in 240 μL of enzyme-linked immunosorbent assay (ELISA) buffer before being assayed using a commercial Cayman CORT ELISA kit according to the manufacturer instructions (Item No. 501320, Cayman Chemical (Ann Arbor, MI, USA, 2016). The concentration of CORT for each sample was calculated using online software from ElisaAnalysis.com (Elisakit.com, Pty Ltd., Melbourne, Australia) using a four-parameter logistic fit, normalised according to the weight of feathers extracted (pg/mg). To eliminate bias, CORT concentration was also performed by investigators not involved in the experiment and, therefore, unaware of the different treatment groups.

### 2.4. Behavioural Responses

#### 2.4.1. Reactivity Test

Reactivity towards human handlers was assessed on 106 days chosen at random over a 6-month period when birds were between 8 and 13 months old, which corresponds to the age range when birds are either selected for slaughter or kept for breeding purposes [23]. The tests were conducted by two handlers (one familiar and one unfamiliar to the birds) who always wore similar clothing to those worn during the treatment periods. The familiar handler had interacted with the birds during the HP1 and HP2 treatments for the first three months while the unfamiliar handler did not. The reactivity test followed the same procedure as described by Bonato et al. [16]. Briefly, within the home pen comprising of 207 birds (2013 and 2015 birds that survived to this age: SAB = 160; ZB = 5; SAB × ZB = 19; SAB × KR = 19; KR = 4; 109 males and 98 females) of mixed sex and treatment, 20 birds per session (10 males and 10 females, irrespective of treatment) were randomly observed and recorded for their specific behaviours towards a handler. The behaviours recorded included approach (bird coming towards the handler), touch (bird could be touched by the handler), wing flapping (bird raising feathers up and down as the handler approached), avoidance (maintaining distance from the handler), excessive pecking (repeatedly grabbing the handler’s body or clothes), and aggression (hissing and/or kicking at the handler). In addition, sexual displays directed towards the handler, such as kantling, stepping, and clucking by males, as well as crouching and clucking by females, were recorded [24,25]. The expression or lack of expression of each of these behavioural traits was recorded in a binomial format as 1 or 0, respectively. Each bird had 1 min of testing, thus one test session lasted for 20 min. Tests were performed by familiar and unfamiliar handlers in a random order; one observation session each day, either in the morning, afternoon, or late afternoon. In total, 51 sessions were performed by the familiar handler and 55 by the unfamiliar handler during the test period of six months. This test was performed at the flock level, as it is standard practice for keeping juvenile ostriches in flocks. This also allowed for the development of a normal social structure before the observations started. It was also assumed that all birds had equal chances to approach and interact with the handlers. Furthermore, although it is conceded that an individual might have followed his counterpart when approaching the human handler, if the former was fearful of the humans, the bird would not necessarily have allowed itself to be touched by the human handler and/or would have kept a distance when the human handler attempted to touch it. Hence, such an interaction was recorded as approach: 1; touch: 0; avoidance: 1 [16].

#### 2.4.2. Docility Test

The docility test was adapted from that described by Mazurek et al. [26] and performed twice on all birds that survived to 12 months of age, which corresponds to the time when ostriches reach puberty and become difficult to handle [27]. A small group of birds (between 15 and 20) was selected from the main flock in a home pen (2013 and 2015: *N* = 123: SAB = 99; ZB = 3; SAB × ZB = 9; SAB × KR = 8; KR = 4; 59 males and 64 females) and moved to a holding pen immediately next to and separated from the test pen (29 m × 35 m) by a fence that allowed visual contact between the birds in the two pens. One bird at a time from the holding pen was caught by two experienced stockmen who were not involved in any stages of the experiment. The bird was then identified through its neck tag, gently guided, and released at the gate of the test pen. If the bird caught had already been tested, it was immediately released within the holding pen; all birds were only tested once. After testing all birds in the holding pen, they were moved to a different pen. A different group of birds was then moved from the home pen to the holding pen for testing. When birds were brought to the test pen, the handler performing the test stood approximately 3 m away from the test pen gate. Each bird was given 10 s to familiarize itself with the test pen before the test started. Either the same familiar or unfamiliar handler involved in the reactivity test then encouraged the bird to enter and remain in a square (8 m × 8 m) drawn on the ground at a corner opposite to the holding pen for 30 s (Figure 1), using slow arm movements and a calm voice. The square was situated far away from the holding pen so as to evaluate whether an individual bird could be guided away from its peers and be contained in the square. Each bird was tested once by the familiar and once by the unfamiliar handler. While the familiar handler was performing the test, the unfamiliar handler was recording the data, or vice versa. The test was terminated if the bird either was contained in the square for 30 s, could not be moved into the marked square within 3 min, or threatened/charged the handler. Latency to enter the marked square and duration of remaining there were recorded. The rationale for this test was that birds taking less time to be encouraged to enter and/or be contained in the marked square for 30 s without showing aggressiveness towards the handler were considered to be docile and less temperamental. On the other hand, those that took more time to be encouraged to enter and/or could not be contained, as well as those showing aggression towards the handler, were considered temperamental and not docile. Defecation and vocalisation were also recorded as they may indicate stress during handling [28,29].

#### 2.4.3. Fear Test

The fear test was adapted from that described by Mazurek et al. [26]. It was performed only on birds from the 2015 batch who reached 12 months of age (*N* = 83: SAB = 65; ZB = 2; SAB × ZB = 6; SAB × KR = 8; KR = 2; 39 males and 44 females). As with the docility test, this test was timed to correspond to the period when the birds reach puberty [27], and was conducted a week after the docility test ended. All birds underwent the fear test twice, once with the same familiar handler and once with the same unfamiliar handler who performed the reactivity and docility tests. While the familiar handler was performing this test, the unfamiliar handler recorded the data, and vice versa. The surface of the test pen (55 m × 29 m) was divided into 18 equal squares (9.4 m × 9.4 m; Figure 2) that were assigned a value according to their distance from the feed trough: 0 for the square containing the feed trough; 1 for adjacent squares; then 2, 3, 4, and 5 for the squares progressively further away (Figure 2). Because of the shortage of workforce in 2013, the fear test could not be performed when these birds were 12 months of age. A small group of birds (15–20 birds) was randomly selected from the main flock and moved into a holding pen adjacent to the test pen in which they had free access to food and fresh water. The fence separating the test and holding pens allowed visual contact between the birds in both pens. One bird at a time was randomly caught in the holding pen by two experienced stockmen not involved in any stages of the experiment. The bird was then identified through its neck tag, gently guided, and released at the gate of the test pen. Each bird was given 10 s to familiarize itself with the pen before the test started. The test lasted 4 min and was composed of three phases. Phase 1 (0–1 min): the bird was left alone in the test pen; phase 2 (1–2 min): the handler entered the test pen, placed food in the feed trough, and then walked out; phase 3 (2–4 min): the handler entered the test pen and stood next to the feed trough and offered food to the bird. The number of squares crossed by the test bird, as well as the position of the bird, was recorded in each phase, in order to allow calculation of the distance travelled as well as the average distance the bird kept from the feed trough. The crossing of fewer squares in each phase (with handler in or not in the camp) reflected calmness, while a higher number of squares crossed in each phase indicated an agitated/temperamental bird. A boundary leading to a square was considered as crossed if the bird placed both feet in it. During the second and third phases, the latency until feeding after the handler put food in the feed trough and while the handler was standing close to the feed trough was recorded. The time taken for each bird to interact with the handler (pecking as an exploratory behavior [30]), or accepting feed from the handler was recorded in phase 3. Birds were offered the same ostrich finisher diet that they received in their home pen [31] and were not fasted prior to the test. Birds that threatened/charged the handler were recorded as being aggressive (in a binomial format: aggressive 1; not aggressive 0), while behavioural stress indicators during the test (similar to those mentioned in the docility test) were also recorded. Each interaction was recorded in a binomial format (1: interacted with and 0: did not interact with the handler).

### 2.5. Statistical Analysis

The H/L ratio before and 72 h after feather harvesting/clipping was compared using a paired *t*-test. Generalized linear mixed models (GLMM) were used to evaluate the effect of treatment, breed, sex, time of sampling, year, and their interactions on the acute stress responses (H/L ratio) for feather harvesting and feather clipping. The basal H/L ratio before sampling was entered as a covariate, while bird identity was included in the model as a random variable to account for repeated pre- and post-sampling records on the same individual. To evaluate the effects of treatment, sex, feather section, and their interactions on the chronic measure of stress, a GLMM was performed by entering CORT concentrations as the dependent variable. The CORT concentration data were log transformed to follow a normal distribution and bird weight was entered as a covariate during the analysis, while bird identity was entered as a random variable to account for repeated sampling of the respective feather segments on the same individual.

The effect of treatment, year, familiarity of handler, sex, breed, and interactions between sex and treatment, as well as year and treatment on behavioural responses, was evaluated using GLMMs. Bird identity was entered as a random variable to account for repeated records on the same individual. Age was also entered as a fixed factor to evaluate the change in behavioural responses over time (between 8–13 months of age). All behavioural responses towards human handlers, except for approach/avoidance of the handler were analysed based on data of birds that approached the handler. As the data on behavioural observations was recorded in binomial format (0 or 1), the logit link function of the GLMM was applied to normalize the data. The data was reported as the logit link function back transformed predicted least square mean estimates.

For the docility test, the time taken by birds to enter, and the period of time the bird was contained within the marked square, were entered as dependent variables in a GLMM, while treatment, year, familiarity of handler, sex, breed, and the interaction between sex and treatment, as well as year and treatment, were entered as fixed factors with bird identity as a random variable to account for repeated records from the same individual. A similar GLMM was performed to evaluate the success of containing the birds in the marked square for 30 s during the docility test.

The fear test was analysed using a GLMM with the number of square areas crossed during each phase, the distance each bird kept away from the stimulus square, as well as time taken to feed or interact with the experimenter, as dependent variables (log transformed). Treatment, sex, breed, and the interaction between treatment and sex were entered as fixed factors, while bird identity was entered as a random variable to account for repeated records on the same individual. A similar GLMM was performed to analyse interactions with the handler following normalization using the logit link function. The probability-value difference (PDIFF) was used for pairwise comparison for the corticosterone analysis because the number of observations sampled was balanced across treatments [32], while the Tukey pairwise comparison test was used for the H/L ratio analysis, as well as the behavioural responses owing to the unbalanced number of observations. Statistical significance was set to *p* < 0.05 and all analyses were performed using SAS, version 9.3 [33].

## 3. Results

### 3.1. Physiological Measures of Stress

There was a significant difference between the H/L ratio before and 72 h after feather harvesting (*t* = −3.18, d*f* = 319, *p* = 0.002; Figure 3a) and feather clipping (*t* = −5.41, d*f* = 121, *p* = 0.001; Figure 3b).

When analysed relative to the three experimental treatments, a significant increase in the H/L ratio was seen after feather harvesting (F_2,319_ = 69.72, *p* < 0.001; Figure 4a) and clipping (F_2,121_ = 7.72, *p* < 0.001; Figure 4b) in the HP2 and S treatments, but not in the HP1 treatment (*p* > 0.05). Furthermore, feather harvesting resulted in a higher (*p* < 0.05) H/L ratio at 72 h after sampling than feather clipping (Feather harvesting: 8.61 ± 1.24 versus feather clipping: 4.67 ± 0.96). While there was no effect of breed, sex, and their interaction on the H/L ratio before and after feather harvesting/clipping, a significant effect of year was observed with a H/L ratio pre- and post-feather harvesting higher in 2013 compared with 2015 (2013: H/L_0_ vs. H/L_72_: 7.61 ± 0.32 vs. 10.84 ± 0.51; 2015: 2.75 ± 0.13 vs. 4.57 ± 0.30; *p* < 0.01).

Feather CORT concentrations were independent of the sections analysed (*p* > 0.05). Mean CORT concentrations were significantly lower in feathers collected from the HP1 compared with the S treatment (F_2,42_ = 4.23, *p* = 0.02; Figure 5), but there were no significant differences between the HP1 versus HP2 and HP2 versus S treatments (*p* > 0.05; Figure 5). There was also no significant effect of any other explanatory variable on feather CORT concentrations (*p* > 0.05).

### 3.2. Behavioural Responses 

#### 3.2.1. Reactivity Tests towards Humans 

There was no effect of treatment or change in behavioural reactivity towards human handlers over time by the birds between 8–13 months of age (*p* > 0.05). However, significant differences were recorded between responses to the familiar and unfamiliar handlers (*p* < 0.05). Specifically, a higher proportion of HP1 and HP2, but not of S birds, were more likely to approach a familiar handler than an unfamiliar handler and preferred to avoid touch from an unfamiliar handler compared with a familiar handler (1.56 ± 1.29 vs. 4.37 ± 1.33, F_1,69_ = 4.44, *p* = 0.04, respectively). Wing flapping was also significantly (*p* < 0.05) more likely to be directed at an unfamiliar as opposed to a familiar handler (Table 1).

There was a significant effect of year on ostrich behaviour (Table 1) and sexual behavioural responses (Table 2) directed towards the handlers. Specifically, birds from the 2015 group were more likely (*p* < 0.05) to hiss and to allow touch by the human, and they stepped and kantled more than those from the 2013 group (*p* < 0.05).

A significant effect of breed was recorded on wing flapping (Figure 6) and hissing (Figure 7) (*p* < 0.05): while wing flapping was similar between SAB and SAB × KR birds, SAB birds flapped more than ZB, SAB × ZB, and KR birds (*p* < 0.05), and SAB × KR birds flapped more than ZB birds (*p* < 0.05). In addition, ZB birds flapped (*p* < 0.05) less than SAB × ZB birds. Birds of the SAB, KR, SAB × KR, and SAB × ZB breed hissed more often than ZB birds (SAB vs. ZB: F_4, 1112_ =6.09, *p* = 0.001; SAB × ZB vs. ZB: F_4,1112_ =3.03, *p* = 0.02; KR vs. ZB: F_4,1112_ = 2.60, *p* = 0.04; Figure 7). However, none of the experimental birds were observed kicking at either human handler, suggesting a lack of physical aggressiveness. Lastly, there were no significant interactions among fixed effects for any of the other traits measured (*p* > 0.05).

#### 3.2.2. Docility and Fear Responses

There were no effects of treatment, familiarity, sex, breed, or their interactions on docility test indicators, stress responses indicators (frequent vocalization, and/or excessive urination, or defecation) and aggressiveness towards the handler during the test (*p* > 0.05). However, a significantly (F_1,212_ = 38.71, *p* = 0.001) greater proportion of the birds from the 2013 cohort (84%) were contained within the marked square during the docility test than those from 2015 (25%). There were no effects of treatment, familiarity, sex, breed, or the interaction between treatment and sex on any of the measures of fear monitored, the average distance kept by birds from the stimulus square during each phase of the test, the number of square boundaries crossed by birds during the test or the time taken by the birds to feed or interact with the handler (*p* > 0.05).

## 4. Discussion

### 4.1. Physiological Measures of Stress

The results of this study revealed that extensive human presence and regular gentle handling of ostrich chicks at an early age (specifically physical interactions) reduced the acute stress response to feather harvesting/clipping and long-term feather CORT concentrations, relative to chicks in which human interactions were limited to essential husbandry tasks. 

The majority of routine farm management practices (i.e., weighing, vaccination, dosing, blood sampling) may be regarded as stressful for animals, as they often need to be physically restrained [7]. Even when procedures are non-invasive, animals not habituated to human handling may be more stressed or suffer fear [34] during such husbandry practices. In this study, feather harvesting and clipping resulted in an elevated H/L ratio 72 h after each procedure was performed. This suggests that these procedures were stressful for the ostriches and the difference in the size of the effect suggests that feather harvesting was more stressful to the birds than feather clipping. As feather harvesting involves pulling-out feather quills from the feather follicles, which might be surrounded by nerve fibres similar to what was found in emus [35], there might also be pain associated with harvesting the feathers in this manner, which could have added to the result. The finding in this study is consistent with the results of Kamau et al. [21], who reported that the H/L ratio in ostriches increased after birds were exposed to a known stressful event such as transportation and may be elevated for up to four days after the event. Regardless of the method of feather collection (harvesting or clipping), ostriches that were raised with an extensive human presence and gentle handling were less affected in terms of the H/L ratio than birds that had limited human–bird interactions at an early age. The potential positive effects of extensive human presence and gentle interactions at an early age on stress responsiveness in ostriches is further supported by the observation that feather CORT concentrations were also significantly lower in this group relative to chicks that had reduced or minimal human contact. As feather CORT concentrations provide an integrative measure of circulating CORT over the time course of the feather growth [22], and, therefore, provide a measure of HPA activity over the previous three-month period, these results suggested that extensive human presence and gentle interactions with ostrich chicks may reduce chronic stress sensitivity caused by routine farm management. These findings are consistent with previous studies where gentle interaction with animals such as tactile stimulation has been reported to have organizational effect on the activation of the HPA axis [36,37,38], and reduce routine stress in a variety of animals including chickens [15,39], lambs [34], dairy cattle [40], and rats [36]. Unlike the H/L ratio test, feather CORT has not yet been employed as an indicator of stress in ostriches. However, based on the application of similar tests in other avian species [22], as well as the reasonable outcome of the present study, it also appears to be an appropriate indicator of long-term stress in ostriches.

### 4.2. Behavioural Responses

#### Reactivity Test

In this study, no significant differences were observed between husbandry treatments on the birds’ willingness to approach or be touched by the handlers, or in aggressiveness towards the handlers. Also, birds’ responses towards the handlers did not change over time between 8–13 months of age. The lack of differences between treatments in this study is inconsistent with previous studies performed on dairy cows [12], pigs [41,42,43], and chickens [15,44], in which a positive relationship was observed between human exposure during development and human approach behaviour later in life. A possible explanation for the lack of an effect of husbandry practices in this study is that chicks from the S treatment may also have associated humans with the provision of feed and fresh water, even when human presence was only limited to routine farm management operations. It is notable that the largest differences between treatment groups in the previous study by Bonato et al. [16] were between pen-reared chicks exposed to extensive human presence compared with chicks reared by foster parents. They accordingly did not report differences in touch interactions between chicks reared using basic husbandry practices (supply of food and water only) and chicks that received some level of human presence. Interestingly, even though there was no effect of early human exposure on approach behaviour, birds exposed to extensive human presence (HP1 and HP2) even without direct gentle handling appeared to be able to differentiate between a familiar and an unfamiliar handler, as revealed by their inclination to approach the familiar handler. The concept that livestock can discriminate between familiar and unfamiliar human handlers has been reported previously (chickens: [45]; pigs: [46,47]; and cattle: [12,48]) and this may benefit animal welfare [9,34]. In this study, the effect of handler was independent of the clothes worn by the handlers, as this was standardised, a result that contrasts with that reported in cattle [48,49]. It was reported that cows were unable to differentiate between people wearing clothing of the same colour, but were capable of distinguishing between people wearing differentially-dyed clothing [48,49]. This observation may indicate that ostriches are not only capable of distinguishing between familiar and unfamiliar handlers, but are able to recognize their handlers by visual markers. Although this is the first study to report this in ostriches, more studies may be required to fully understand how ostriches use cues to discriminate not only between handlers, but also between objects and/or edible/non-edible food, as ostriches in the wild spend most of their time selectively foraging for nutritious food [50]. Furthermore, the inclination of birds to associate with a familiar human handler can be interpreted as a lack of fear for that handler and may indicate that the presence of a familiar human during handling would impact less on the welfare of an animal. Sexual behaviour of ostriches towards human handlers in this study was, however, not influenced by husbandry treatment or any of the other factors evaluated. Although ostrich sexual behaviour is known to be stimulated by human presence mostly in birds reared artificially by humans (because human–ostrich bonds are alleged to be formed during the rearing period; [24]), the lack of differences between treatments may be attributed to the relatively young age of the birds in this study when the behavioural observations were performed (8–13 months old). Further investigations on sexual display towards humans would thus be needed when the birds reach sexual maturity at an age of 2–3 years for clarification. 

### 4.3. Docility and Fear Responses

No effect of husbandry treatment on ostrich temperament at puberty was found during the docility and fear tests in this study. Although it was evident that early human–animal interactions affected the activation of the HPA axis, as shown at 7.5 months of age, it did not affect the behavioural responses and temperament of birds at 12 months of age. This lack of effect on ostrich docility and fear may be explained in several ways. First, the effect of early human interactions may not persist over the post treatment–pre-test period from 3–12 months. Secondly, any effect may have been over-ridden by the stress of the procedure immediately prior to the test, namely being caught by an unfamiliar catching crew. Thirdly, as the tests used were adapted from those used previously with cattle, ostriches might have experienced too high levels of stress during the test to allow the effect of treatment on temperament and ease of handling to be observed. However, behavioural responses such as frequent vocalization and/or excessive urination or defecation, which typically increase in stressed and/or agitated animals [7,28], including ostriches [29], were not observed during the tests in this study, suggesting that the birds were not stressed/agitated by the test. Lastly, as the birds were subjected to behavioural observations between 8–13 months old, they might have become habituated to human presence and/or associated humans in general with the provision of food and water, which in turn could have overridden any effects of early-life experiences on temperament at a later stage in life. The effects of early human–animal interactions versus later human–animal interactions on juvenile ostrich temperament as assessed in the current study clearly need more investigation. Future research should consider two methodological aspects of the current study in particular. The first is in refining the docility and fear tests to make them more suitable to ostriches. The second is to be more rigorous in the exposure birds have to humans, although this could be difficult to achieve within a commercial set-up. In addition to this, in the current study, the birds were repeatedly tested by the same ‘unfamiliar’ handler, which could have compromised the definition of ‘unfamiliar’ and explain the lack of differences between the familiar and unfamiliar handlers during the docility and fear tests. Studies evaluating the flight speed and distance covered by birds after handling as described by Burrow et al. [51] may provide a more accurate estimate of docility and fear responses in ostriches. This is because docile and less fearful animals have been observed to vacate the handling/weighing procedure at a much slower pace than temperamental animals [51]. Such studies could also assist in the selection of docile birds that show less fear of humans during routine husbandry practices (e.g., tagging, weighing).

Finally, a higher success rate for the docility test was obtained in the 2013 birds compared with the 2015 birds. Birds hatched in 2013 were also contained for a longer period of time in the marked square compared with the 2015 birds. This may be attributed to the level of experience of the handlers, which was greater in 2013 than in 2015. Previous studies have reported that more experienced stockmen handle animals in a manner that results in lower stress responses [52], and that this can increase productivity and ultimately the welfare of animals [14,52,53].

## 5. Conclusions

The results of this study demonstrated the benefits of extensive human presence and gentle handling during the three months post hatch period on measures of physiological stress responses and reactivity towards a familiar and an unfamiliar handler in farmed ostriches. Ostrich chicks exposed to extensive human presence—including touch at an early age—had reduced physiological stress sensitivity later in life compared with chicks exposed to limited human presence and interactions. This highlights the importance of early positive human–animal interactions in improving the welfare of ostriches by reducing physiological stress sensitivity later in life. Behavioural responses of juvenile ostriches towards humans were not influenced by early husbandry treatment, but birds exposed to extensive human presence as chicks were able to distinguish between a familiar and an unfamiliar handler and adjust their behaviour accordingly. The potential effect of early positive human–animal interactions on ostrich chicks appears to have influenced the activity of the HPA axis in this study. Also, the lack of treatment effects in the behavioural testing of juvenile ostriches may indicate that early treatment effects might not affect ostrich temperament later in life. It might alternatively be that the frequency of human–animal interactions during the repeated behavioural (reactivity) testing over-shadowed the early treatment effects and thus influenced the results of the later docility and fear tests. That notwithstanding, the data indicate a need for further research into the effects of early treatment on the activity of the HPA axis at 12 months, and the need to refine behaviour testing for this species. This will facilitate improved welfare, and potentially improve selection of docile animals for breeding stock.

## Figures and Tables

**Figure 1 animals-08-00175-f001:**
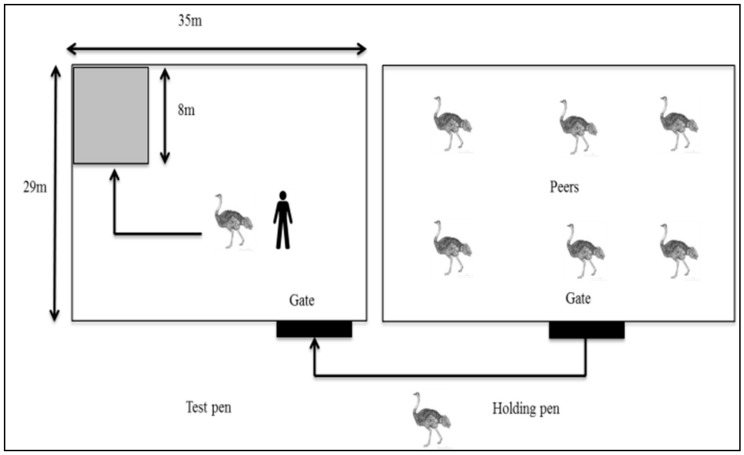
Experimental procedure used during the docility test showing the holding pen (with peers) and the test pen (with a marked square drawn on the corner opposite the gate). One bird was taken from the holding pen to the test pen and encouraged to enter the marked square and remain contained there for 30 s by either a familiar or an unfamiliar handler.

**Figure 2 animals-08-00175-f002:**
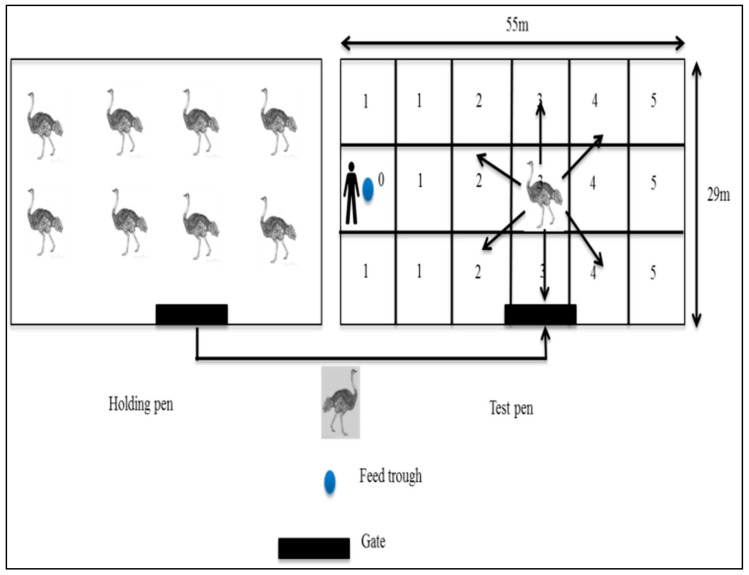
Experimental procedure of the fear test showing the holding pen (with the peers) and the test pen (divided into 18 equal squares). One bird was randomly selected from the holding pen and taken to the test pen and its movement was recorded during three different phases. Phase 1: 1 min; no human present in the test pen. Phase 2: 1 min; human introduced food and walk out of the test pen. Phase 3: 2 min; food and human present in the test pen (human offering food to the bird).

**Figure 3 animals-08-00175-f003:**
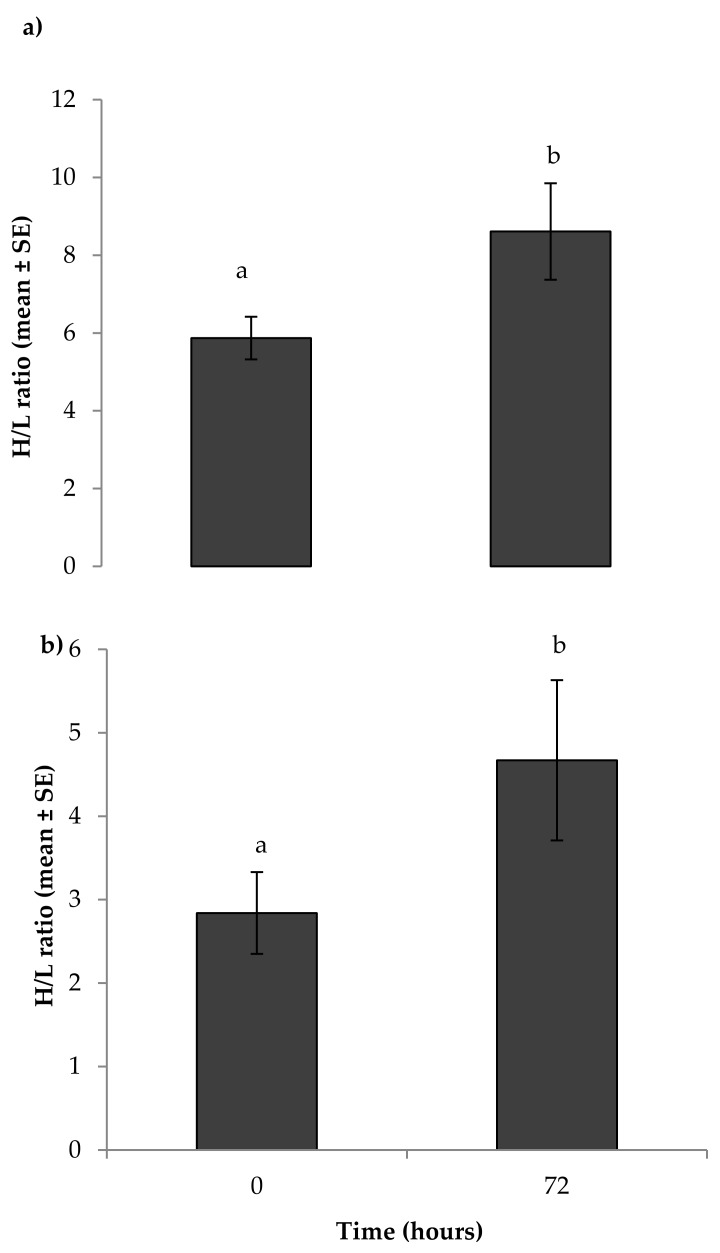
Mean heterophil/lymphocyte (H/L) ratios as a measure of acute stress responses of 7.5 months old ostriches before and 72 h after: (**a**) feather harvesting (*N* = 238); (**b**) feather clipping (*N* = 87). Means with different superscripts differed significantly (*p* < 0.05).

**Figure 4 animals-08-00175-f004:**
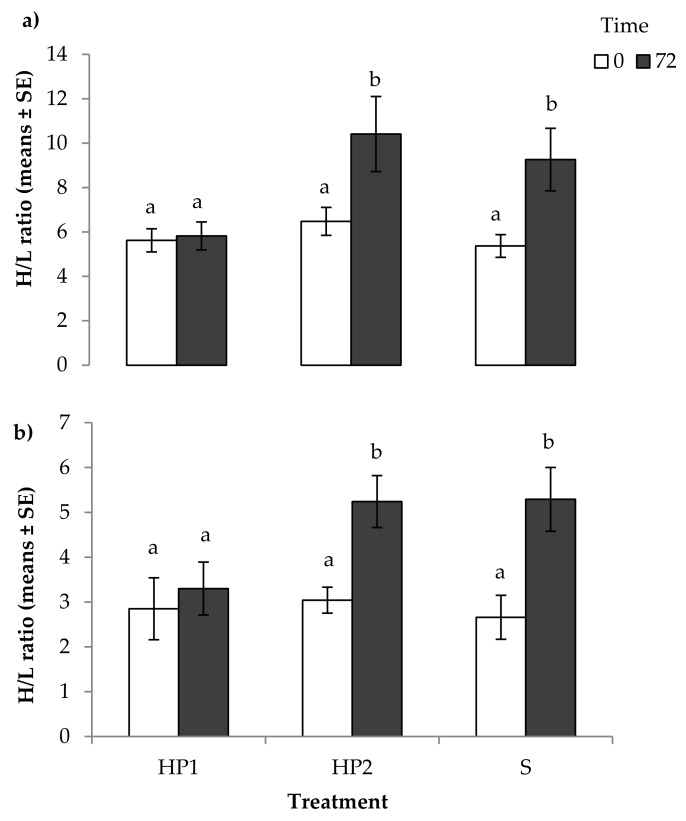
Mean heterophil/lymphocyte (H/L) ratios of the interaction between husbandry treatment and sampling time as a measure of acute stress responses of 7.5-month-old ostriches exposed to husbandry treatments that differed in the extent of human presence and interactions with the birds before and 72 h after: (**a**) feather harvesting (*N* = 238); (**b**) feather clipping (*N* = 87). For feather harvesting: the birds in Human Presence 1 treatment (HP1, *N* = 74) were exposed to extensive/prolonged human presence with physical contact (touch, stroking), gentle human voice, and visual contact. The birds in HP2 treatment (*N* = 87) were exposed to extensive human presence with gentle human voice and visual contact, but without physical contact. The birds in S treatment (*N* = 77) had human presence and visual human contact limited to the provision of food and fresh water. For feather clipping: HP1 treatment (*N* = 28), HP2 treatment (*N* = 29), S treatment (30). Means with different superscripts differed significantly (*p* < 0.001).

**Figure 5 animals-08-00175-f005:**
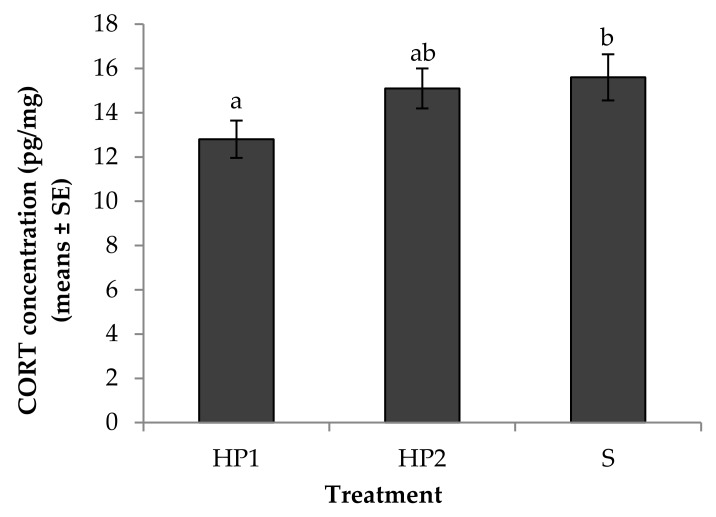
Means depicting the effect of human exposure of ostrich chicks for three months on their corticosterone (CORT) concentration from the floss feathers harvested at 7.5 months old as an indicator of chronic stress responses. (*N* = 48). The birds in HP1 treatment (*N* = 16) were exposed to extensive human presence with physical contact (touch, stroking), gentle human voice, and visual stimuli. The birds in HP2 treatment (*N* = 16) were exposed to extensive human presence with gentle human voice and visual stimuli, but without physical contact. The birds in S treatment (*N* = 16) had human presence limited to the provision of food and fresh water. Means with different superscripts differed significantly (*p* < 0.05).

**Figure 6 animals-08-00175-f006:**
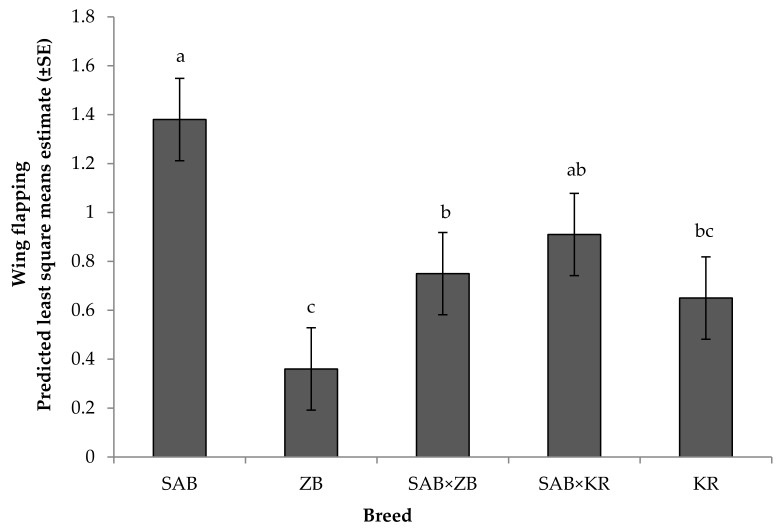
Means depicting the effect of breed on wing flapping by ostriches in the reactivity tests that approached the handler (*N* = 207). SAB: South African Black (*N* = 160); ZB: Zimbabwean Blue (*N* = 5); KR: Kenyan Redneck (*N* = 4); and their cross-bred combinations: SAB × ZB (*N* = 19) and SAB × KR (*N* = 19). Means with different superscripts differed significantly (*p* < 0.05).

**Figure 7 animals-08-00175-f007:**
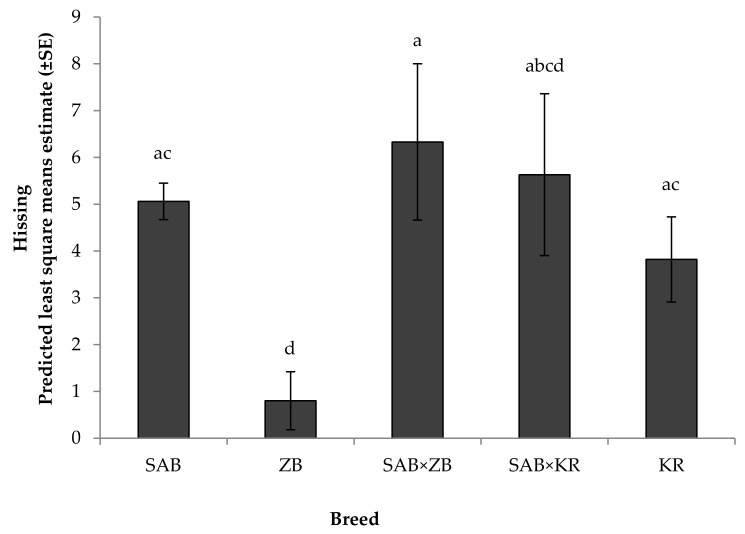
Means depicting the effect of breed on hissing by ostriches in the reactivity tests that approached the handler (*N* = 207). SAB: South African Black (*N* = 160); ZB: Zimbabwean Blue (*N* = 5); KR: Kenyan Redneck (*N* = 4); and their cross-bred combinations: SAB × ZB (*N* = 19) and SAB × KR (*N* = 19). Means with different superscripts differed significantly (*p* < 0.05).

**Table 1 animals-08-00175-t001:** Predicted least square mean estimates (±SE; derived from back transformation of the logit function) from reactivity tests of handler familiarity, experimental year, and sex on ostrich behavioural activities as directed towards the human handlers.

Effects	Behavioural Responses				
	Approach	Touch	Wing Flapping	Excessive Pecking	Hissing
Familiarity:					
Familiar	3.95 ± 0.45	2.87 ± 0.41	0.38 ± 0.20	2.23 ± 0.36	4.29 ± 1.02
Unfamiliar	1.99 ± 0.36	2.07 ± 0.37	0.96 ± 0.21	2.30 ± 0.37	4.36 ± 1.05
*p*-value	<0.01	>0.05	<0.05	>0.05	>0.05
Year:					
2013	3.34 ± 0.45	2.05 ± 0.34	0.61 ± 0.19	2.10 ± 0.34	3.34 ± 0.62
2015	2.59 ± 0.33	2.89 ± 0.33	0.72 ± 0.16	2.42 ± 0.33	5.32 ± 0.79
*p*-value	>0.05	<0.05	>0.05	>0.05	<0.05
Sex:					
Male	2.69 ± 0.30	2.50 ± 0.31	0.76 ± 0.17	2.63 ± 0.33	4.42 ± 0.60
Female	3.25 ± 0.37	2.44 ± 0.31	0.58 ± 0.17	1.90 ± 0.32	4.23 ± 0.61
*p*-value	<0.05	>0.05	>0.05	<0.01	>0.05

**Table 2 animals-08-00175-t002:** Predicted least square mean estimates (±SE; derived from back transformation the logit function) from reactivity tests for sexual behavioural responses of ostriches relative to the year of observation (*N* = 207).

Effect	Sexual Behavioural Responses				
	Males			Females	
Year	Clucking	Stepping	Kantling	Clucking	Crouching
2013	6.19 ± 5.38	1.69 ± 0.99	3.66 ± 0.81	6.29 ± 16.47	6.60 ± 8.86
2015	12.5 ± 7.20	4.54 ± 1.13	5.29 ± 0.74	11.60 ± 16.83	9.29 ± 8.77
*p*-value	>0.05	<0.01	<0.05	>0.05	>0.05

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
