# Peer review of "The Effect of Extensive Human Presence at an Early Age on Stress Responses and Reactivity of Juvenile Ostriches towards Humans"

_animals, 2018, doi:10.3390/ani8100175_

Round 1

Reviewer 1 Report

The research topic is very interesting and the topic is important to ostrich welfare and thus within the scope of the journal. However, I am afraid the study design has serious flaws and additional tests would have been necessary.

Treatment effects were only found for physiological, but not for behavioural data. It cannot be ruled out that this was due to the study design. The treatments were performed until the age of 3 months. The physiological parameters were collected at the age of 7.5 months. Reactivity tests towards humans were performed weekly from 8 to 13 months, but it is not clear how often how many animals per treatment were tested. Besides the tests/observations were performed in the flock (i.e. at group level) but analysed at an individual level. However, the birds might have influenced each other’s’ reactions. Thus the meaningfulness of these tests is questionable. The other behavioural tests - that were performed at an individual level - were performed when the birds were already 12 months old. Thus, actually, a potential treatment effect on physiological and behavioural data cannot be really compared due to the time lag. It would have been necessary to perform individual behavioural tests for the assessment of the human-animal relationship closer in time to the physiological data collection. Besides the docility test that was used is not a classical test to assess the human-animal relationship in cattle (as e.g. approach or avoidance distance tests). It is rather used for assessing the docility, which is also influenced by genetic factors, see temperament testing. It was originally developed for breeding selection of bulls according to their dangerousness/ease of handling. Bulls scored as aggressive in the docility test have partly low avoidance distances, i.e. the measures are not always correlated.

Overall, since no differences in the behavioural measures were found, there is no proof that the treatment was effective in improving the human-animal relationship, or reducing fear of humans. Due to the study design it is not possible to tell if there was no treatment effect or if there might have been a shorter lasting effect (reflected in the physiological data), which could not be measured since the behavioural tests at an individual level were performed later. The tests/reactivity observations at flock level do maybe not deliver meaningful results because they were performed 1) at herd level and 2) the data of the weekly tests over the months 8 to 13 were pooled.

Even if the treatment might not have been effective to change the human-animal relationship, it might have had an impact on the HPA axis via organisatorial effects on the activity of the HPA axis. The physiology results are in line with findings that tactile stimulation might have an organisatorial effect on the activity of the HPA axis (see e.g., Schulz et al., 2007; Lin et al., 1997; Lürzel et al., 2015). Thus the effect on physiology cannot be only explained by an improved human-animal relationship. Thus there might have been really no treatment effect on the human-animal relationship but still an effect on the HPA axis. Or a potential effect on the human-animal relationship did not persist until the age of 12 months due to the handling experiences in later life.

Maybe parts of the study (physiological data) could be still published as short communication, but not as the present manuscript “The effect of extensive human presence at an early age on stress responses and reactivity of juvenile ostriches towards humans”.

Regarding the scientific merit it would be best if parts of the study could be repeated, with individual behavioural tests (such as avoidance or approach tests, and again the "fear test") and physiological measures around the age of 7.5 months.

The behavioural tests should be performed prior to the potentially stressful handling procedures during which physiologicaly data were collected.

Further comments:

Some parts do not have sufficient information for the study to be replicable. For instance, how many animals per treatment/breed/gender were subjected to which behavioural test? More information on the husbandry during the treatment and testing as well as information on the treatments itself would be needed.

I would advise to be more careful in the wording and use e.g. “stress-related parameters” instead of e.g. “measurements of physiological stress responses”, or just “measurements of physiological stress responses”; corticosterone levels are not only altered by stress, but for instance also by general activity

Abstract:

Ln 33 N=13 should be N=136?

Ln 47-48: “Birds’ 47 reactivity towards humans was not affected by treatment (P>0.05)” and ln 50-53 “The results indicate that early gentle human interactions with ostrich chicks are beneficial …..promote a positive human-animal relationship as demonstrated by the inclination of birds to associate with a familiar handler.” are contracting each other. Besides the results do not support the conclusion regarding the promotion of a positive human-animal relationship

Introduction: ln 65-66: the links between a positive human-animal interactions and cortisol or corticosterone concentrations are not always so clear; cortisol or corticosterone concentrations are also affected by other factors, e.g. activity

Material & methods:

Ln 93: information about the rearing facility, and type of housing during the treatments should be added;

Ln 99-110: information about the group composition, group size, housing should be added;

Did the treatment groups (HP 1, HP 2 and S) have visual contact?

How many animals were in a treatment group at the same time?

Were all animals per group subjected to the respective treatment, e.g., were all animals stroked? Could all animals be stroked/touched?  

How many hours of treatment (HP1 and HP2) did the animals receive in total until the age of 3 months? And how much time per day did humans spend approximately in the S group?

Who performed the treatment (amount of people, gender, body height)

Ln 125: was the person counting the blood cells blinded?

2.4 Behavioural responses: who recorded the behavior?

2.4.1 Reactivity towards humans

Information on the handlers is needed; was the unfamiliar/familiar handler always the same for all birds?

What was the test order? E.g., first familiar, then unfamiliar handler ? How often did each person perform the test?

The duration of the test should be mentioned. The cited reference might not be available to all readers.

The birds’ behaviour was observed within the flock, thus individual birds’ behavior might have been affected by other birds, e.g., via social facilitation; if one bird approaches others are more likely to approach. Although critically discussed in Bonato et al., 2013, it is not discussed in this manuscript.

From the description it is not clear, if 20 birds per test session were tested or 20 birds per treatment per test session. The information is missing how many birds per treatment group were observed and how often each bird was observed.

How were the 20 birds selected?

2.4.2 Docility test

How was the random selection achieved? Were they preselected via their tags? If they were selected while being already within the flock, more fearful animals might have been hiding.

2.4.3 Fear test

Information on the test order (familiar/unfamiliar) is missing; was the fear test performed before/after the docility test? How much time passed between tests?

How was the random selection achieved? What type of food was fed? Were the animals hungry? Was it a special food?

Ln 203: “time taken for the bird to feed” – is this referring to the latency until feeding or the duration of feeding?

Ln 217-218: was it a separate model for harvesting and a separate model for feather clipping?

Statistics:

Ln 226: “All behavioural responses towards human handlers, except for approach/avoiding the handler were analysed based on data of birds that approached the handler” This might have biased the results and decreased the variability

H/L ratio analyses: Why was the pre-test measure not used as a covariate? It reads as if the pre-test measures were entered as dependent variables.

 Results:

Ln 259: statistical test data should be added in addition to P-value

Results:

The descriptive data for the behavioural measures are not clearly presented. E.g.,

what are the descriptive data standing for in Tab. 1.?

It was indicated that “The expression or lack of expression of each of these behavioural traits was recorded in a binomial format as 1 or 0, respectively”. How does it lead to the presented least square mean estimates?

Or in ln 300: (1.56….).. Is this a measure for the distance in m?

Discussion:

I missed a critical discussion of the docility test, which is not usually used as a measure of the human-animal relationship but more as a temperament measure

Ln 426: “conditioning of the birds as a result of habituation”… how is this meant? Conditioning is learning based on association, in contrast to habituation, which is a type of non-associative learning

The physiology results are in line with findings that tactile stimulation might have an organisatorial effect on the activity of the HPA axis (see e.g., Schulz et al., 2007; Lin et al., 1997; Lürzel et al., 2015). Thus there might have been really no treatment effect on the human-animal relationship but still an effect on the HPA axis. Or a potential effect on the human-animal relationship did not persist until the age of 12 months due to the handling experiences in later life.

Conclusions:

Are not justified regarding the reactivity to humans

Author Response

Comments and Suggestions for Authors

The research topic is very interesting and the topic is important to ostrich welfare and thus within the scope of the journal. However, I am afraid the study design has serious flaws and additional tests would have been necessary.

Thank you for the quality of your report, which we feel have helped us to improve the value of our study. Responses to your comments are in bold below, and in red within the manuscript.

Treatment effects were only found for physiological, but not for behavioural data. It cannot be ruled out that this was due to the study design. The treatments were performed until the age of 3 months. The physiological parameters were collected at the age of 7.5 months. Besides the tests/observations were performed in the flock (i.e. at group level) but analysed at an individual level. However, the birds might have influenced each other’s’ reactions. Thus the meaningfulness of these tests is questionable.

The treatment (i.e. human animal interactions) until 3 months of age is during a crucial imprinting phase that is known to impact behaviour/docility later in life. The different time points used for each variable (physiological and behavioural tests) correspond to different periods of their life where ostriches have to go through specific management practices, according to current practices in the ostrich industry, when alternations in docility might prove beneficial. For instance, feather harvesting and/or clipping are performed when the birds are in a transition from chick feathers to juvenile feathers. The feathers are thus fully mature and ready to be harvested. The timing of behavioural responses also corresponds to a period of time where the birds are selected for either breeding purposes or slaughter. Finally, the timing of the docility and fear tests corresponds to the period of time when birds reach puberty and become difficult to handle. We have clarified the reason for choosing these specific points of time in the text, as well as provided additional references in support (line 146 to 149; line 189 to 191; line 219 to 220 and; line 251 to 253).

The information on birds that may have influenced each other to approach the handler has been clarified and how the data of potential social facilitation was recorded (line 209 to 216). Birds that might have approached the handler as a result of imitating contemporaries were expected to avoid a handler from touching them as a result of fear. In recording behaviour, these birds approached the handler but would not allow to be touched since they are likely to avoid interactions (hence were recorded as: approach: 1; touch: 0; avoidance: 1).

Reactivity tests towards humans were performed weekly from 8 to 13 months, but it is not clear how often how many animals per treatment were tested.

We have clarified this in the text (line 195-199).

The other behavioural tests - that were performed at an individual level - were performed when the birds were already 12 months old. Thus, actually, a potential treatment effect on physiological and behavioural data cannot be really compared due to the time lag. It would have been necessary to perform individual behavioural tests for the assessment of the human-animal relationship closer in time to the physiological data collection.

It was not our intention to directly correlate physiological and behavioural data but  to rather investigate a potential long-lasting effect of habituation to humans done at an early age on specific variables that are considered critical for ostrich farming (i.e. stress related issue with management practices, whether it be feather collection or handling when birds reach the juvenile stage). For example, feather corticosterone levels represent the mean over a long period (weeks to months) rather than at the point of collection. We, thus, clarified the aim of the study (line 86 to 90) and clarified why we performed these tests at the different time points (line 146 to 149; line 189 to 191; line 219 to 220 and; line 251 to 253).

Besides the docility test that was used is not a classical test to assess the human-animal relationship in cattle (as e.g. approach or avoidance distance tests). It is rather used for assessing the docility, which is also influenced by genetic factors, see temperament testing. It was originally developed for breeding selection of bulls according to their dangerousness/ease of handling. Bulls scored as aggressive in the docility test have partly low avoidance distances, i.e. the measures are not always correlated.

This dualism was clarified as suggested to avoid confusion (line 236 to 240). Breed effect was assessed but no effect was recorded.

Overall, since no differences in the behavioural measures were found, there is no proof that the treatment was effective in improving the human-animal relationship, or reducing fear of humans. Due to the study design it is not possible to tell if there was no treatment effect or if there might have been a shorter lasting effect (reflected in the physiological data), which could not be measured since the behavioural tests at an individual level were performed later. The tests/reactivity observations at flock level do maybe not deliver meaningful results because they were performed 1) at herd level and 2) the data of the weekly tests over the months 8 to 13 were pooled.

We clarified in the manuscript why the reactivity test was performed at the flock level (line 208 to 209). We also re-analyzed the data from 8-13 months of age to investigate whether behavioural change could occur over time towards humans by including age in the analysis (line 304 to 305). However no such effect was detected (line 374 to 375).

Even if the treatment might not have been effective to change the human-animal relationship, it might have had an impact on the HPA axis via organisatorial effects on the activity of the HPA axis. The physiology results are in line with findings that tactile stimulation might have an organisatorial effect on the activity of the HPA axis (see e.g., Schulz et al., 2007; Lin et al., 1997; Lürzel et al., 2015). Thus the effect on physiology cannot be only explained by an improved human-animal relationship. Thus there might have been really no treatment effect on the human-animal relationship but still an effect on the HPA axis. Or a potential effect on the human-animal relationship did not persist until the age of 12 months due to the handling experiences in later life.

This was clarified in the manuscript (line 444 to 447; line 459-461).

Maybe parts of the study (physiological data) could be still published as short communication, but not as the present manuscript “The effect of extensive human presence at an early age on stress responses and reactivity of juvenile ostriches towards humans”.

Regarding the scientific merit it would be best if parts of the study could be repeated, with individual behavioural tests (such as avoidance or approach tests, and again the "fear test") and physiological measures around the age of 7.5 months.

This was clarified in the manuscript (as mentioned previously). More specifically, the docility and fear tests performed at 7.5 months of age may not have been accurate to predict ostrich behaviour when they reach puberty and become problematic to handle (which was our main point of interest). We acknowledge that performing individual behavioural tests (such as avoidance or approach tests) would have contribute greatly to this study, but our facility and workforce on the farm did not allow us to conduct such tests. However, we believe that the behavioural tests we conducted do add some values to to the very scare literature available on ostrich behaviour.

The behavioural tests should be performed prior to the potentially stressful handling procedures during which physiologicaly data were collected.

We have clarified why we started with the physiological stress response tests followed by the behavioural tests (line 146 to 149; line 189 to 191; line 219 to 220 and; line 251 to 253). As previously mentioned, performing behavioural tests at an earlier age (7.5 months or before) may not have been accurate enough to predict the behaviour in ostriches when they reach the puberty stage which is where handling difficulties are experienced and aggressive behaviour starts to be exhibited.

Further comments:

Some parts do not have sufficient information for the study to be replicable. For instance, how many animals per treatment/breed/gender were subjected to which behavioural test? More information on the husbandry during the treatment and testing as well as information on the treatments itself would be needed.

The number of animals for each category was clarified throughout the methodology section with appropriate references detailing the husbandry procedures, testing and treatments (physiology tests: line 145; line 165 to 166; reactivity tests: line 194 to 198; docility test: line 219to 221; fear test: line 250 to 255).

 I would advise to be more careful in the wording and use e.g. “stress-related parameters” instead of e.g. “measurements of physiological stress responses”, or just “measurements of physiological stress responses”; corticosterone levels are not only altered by stress, but for instance also by general activity

This was clarified throughout the methodology section. The wording was also revised as suggested.

 Abstract:

Ln 33 N=13 should be N=136?

This was rectified to N=136 (line 35).

Ln 47-48: “Birds’ reactivity towards humans was not affected by treatment (P>0.05)” and ln 50-53 “The results indicate that early gentle human interactions with ostrich chicks are beneficial …..promote a positive human-animal relationship as demonstrated by the inclination of birds to associate with a familiar handler.” are contracting each other. Besides the results do not support the conclusion regarding the promotion of a positive human-animal relationship.

 This was clarified (line 55).

Introduction: ln 65-66: the links between a positive human-animal interactions and cortisol or corticosterone concentrations are not always so clear; cortisol or corticosterone concentrations are also affected by other factors, e.g. activity

This was clarified (line 67 to 73).

Material & methods:

Ln 93: information about the rearing facility, and type of housing during the treatments should be added;

This was added as suggested (line 119 to 126).

Ln 99-110: information about the group composition, group size, housing should be added;

This was added as suggested (line 124 to 130).

Did the treatment groups (HP 1, HP 2 and S) have visual contact? How many animals were in a treatment group at the same time?

This was added and clarified as suggested (line 111 to 118).

Were all animals per group subjected to the respective treatment, e.g., were all animals stroked? Could all animals be stroked/touched?

This was clarified as suggested (line 118 to 122). Chicks from the HP1 treatment were stroked and touched whenever they came close to the handler and allowed touching. The main idea was to try to interact with the chicks in a positive manner with minimal stress, thus chicks were not forced to interact with the handler as this could have induced stress. We added information in the manuscript to explain the difference between the HP1 and HP2 treatments.

How many hours of treatment (HP1 and HP2) did the animals receive in total until the age of 3 months? And how much time per day did humans spend approximately in the S group?

This was clarified as suggested (line 137 to 140). HP1 and HP2 chicks received 343 hours of human presence in the 3 months of the experiment. Assuming that the supply of feed and water in the S treatment group was 30 minutes/day; then S chicks received approximately 48 hours of human visual contact during the 3 months period.

Who performed the treatment (amount of people, gender, body height)

This was clarified as suggested (line 122 to 124). The height of the people differed – some were short and some tall, but similar overalls were worn by each.

Ln 125: was the person counting the blood cells blinded?

We used investigators that were not involved in the experiment to do the counting of the blood cells, and they were, therefore, unaware of the different treatments. We have clarified this, as suggested (line 160 to 161 and line 184 to 185).

2.4 Behavioural responses: who recorded the behavior?

The familiar and unfamiliar handler recorded the behaviour.  This was clarified in the text (Reactivity test: line 190-193), Docility and fear test: the familiar handler recorded the behaviour while the unfamiliar handler was performing the test and vice versa (line 242, line 282).

2.4.1 Reactivity towards humans

Information on the handlers is needed; was the unfamiliar/familiar handler always the same for all birds?

This was clarified in the text.  The same handlers (familiar and unfamiliar) were used for all behavioural tests (line 190-193; line 227 to 228; line 253 to 254).

What was the test order? E.g., first familiar, then unfamiliar handler ? How often did each person

This was clarified in the text (line 206 to 208). The test order for this test was random, but both handlers performed approximately equal number of sessions (N=51 for familiar handler and 55 for unfamiliar handler).

The duration of the test should be mentioned. The cited reference might not be available to all readers.

The duration of the test was added (line 205 to 206). Each bird was tested for 1 minute, thus the test session lasted for at least 20 minutes as 20 birds were tested during a session.

The birds’ behaviour was observed within the flock, thus individual birds’ behavior might have been affected by other birds, e.g., via social facilitation; if one bird approaches others are more likely to approach. Although critically discussed in Bonato et al., 2013, it is not discussed in this manuscript.

The information has been added and how the data of potential social facilitation was recorded (line 208 to 215). Birds that might have approached the handler as a result of imitating contemporaries were expected to avoid a handler from touching them as a result of fear. In recording behaviour, these birds approached the handler but would not allow touching since they are likely to avoid interactions (hence were recorded as: approach: 1; touch: 0; avoidance: 1).

From the description it is not clear, if 20 birds per test session were tested or 20 birds per treatment per test session. The information is missing how many birds per treatment group were observed and how often each bird was observed.

The information on how many birds per test session was added in the manuscript (line 196 to 198). Twenty birds per test session regardless of treatment were tested, each bird had 1 minute to allow for evaluation of its behaviour towards the handler (line 205 to 206).

How were the 20 birds selected?

The information on how the 20 birds were selected was added in the manuscript (line 196 to 198). Twenty birds in visual sight of the handler were observed for their behaviours towards the handler and recorded.

 2.4.2 Docility test

How was the random selection achieved? Were they preselected via their tags? If they were selected while being already within the flock, more fearful animals might have been hiding.

This was clarified in the text (Line 217 to 218; and line 223 to 225). All birds maintained in the home pen were tested. Two stockmen not involved in any of stages of the experiment caught an individual bird at random. The bird was then identified via its neck tag, guided and then released into the test pen by these two stockmen until all birds were tested.

 2.4.3 Fear test

Information on the test order (familiar/unfamiliar) is missing; was the fear test performed before/after the docility test? How much time passed between tests?

This was clarified in the text (line 254 to 255). The fear test was performed a week after the docility test. Each bird was tested first by a familiar and then by an unfamiliar handler.

How was the random selection achieved?

This was clarified in the text (Line 248 to 250). The same procedure than for the docility test was used: all birds that survived to 12 months of age were tested and randomly caught for the test before being identified and tested.

What type of food was fed? Were the animals hungry? Was it a special food?

This was clarified in the text (line 262 to 264). The same food as in the home pen was used, and the birds were not fasted before the test (line 277 to 279).

Ln 203: “time taken for the bird to feed” – is this referring to the latency until feeding or the duration of feeding?

This was clarified in the text as the latency until feeding (line 274 to 275).

Ln 217-218: was it a separate model for harvesting and a separate model for feather clipping?

Feather harvesting and clipping had their own models only for the t-test analysis. They were in the same model for the GLMMs (line 290 to 293).

Statistics:

Ln 226: “All behavioural responses towards human handlers, except for approach/avoiding the handler were analysed based on data of birds that approached the handler” This might have biased the results and decreased the variability.

The behavioural response was considered exhibited when directed specifically towards the handler (in front of the handler). For instance, a bird performing the kantling display away from the handler was considered as not directing it towards the handler. Hence a bird had to approach a handler to exhibit a specific response to be recorded, while for the approaching behaviour, the bird either did or did not approach the handler. Although it is acknowledged that fearful animal (that which might have not approached the handler or their contemporaries towards humans) would be less likely to be recorded, such birds were scored for only approaching human handler or not approaching only.

 H/L ratio analyses: Why was the pre-test measure not used as a covariate? It reads as if the pre-test measures were entered as dependent variables.

This was clarified in the text (line 293 to 294). The pre-test measure was indeed used as a covariate and was mistakenly omitted during the description of the model.

Results:

Ln 259: statistical test data should be added in addition to P-value

This was added as suggested (line 337 to 338).

 Results:

The descriptive data for the behavioural measures are not clearly presented, e.g.,

What are the descriptive data standing for in Tab. 1.?

It was indicated that “The expression or lack of expression of each of these behavioural traits was recorded in a binomial format as 1 or 0, respectively”. How does it lead to the presented least square mean estimates?

This was clarified in the text and in the caption of the table (line 308 to 309; line 381 to 382). The least square means estimates were predicted from the logit link transformation which was used to transform the binomial data for analysis.

Or in ln 300: (1.56….).. Is this a measure for the distance in m?

This was clarified in the text (line 308 to 309).

 Discussion:

I missed a critical discussion of the docility test, which is not usually used as a measure of the human-animal relationship but more as a temperament measure

This was clarified in the manuscript (line 493 to 497; line 500 to 511). The discussion on the docility test as a measure of temperament has been revised in the text for temperament evaluation.

 Ln 426: “conditioning of the birds as a result of habituation”… how is this meant? Conditioning is learning based on association, in contrast to habituation, which is a type of non-associative learning

We have reworded this argument to explain learning based on association (line 506 to 508).

Conclusions:

Are not justified regarding the reactivity to humans

The conclusions has been revised and justified accordingly (line 535 to 542).

Reviewer 2 Report

Overall this is a nice paper. I have some comments specific to text as attached. There is some more work needed on methodological aspects. I have given some thoughts for discussion, including need to be more constructively critical of their own work and how potential design flaws can be addressed in future, notable regarding role of classical learning and what is an 'unfamiliar' person in testing.

Author Response

Overall this is a nice paper. I have some comments specific to text as attached. There is some more work needed on methodological aspects. I have given some thoughts for discussion, including need to be more constructively critical of their own work and how potential design flaws can be addressed in future, notable regarding role of classical learning and what is an 'unfamiliar' person in testing.

Please find our responses to your comments in blue within the manuscript.

Line 17: regarding early handling / experience, it is not an issue of domestication but of socialisation and sensitive periods. Talking of domestication is mis-leading in this regard. The importance of appropriate, pleasant experiences is as true for long domesticated species such as the dog.

This was deleted to avoid confusion (line 18 to 19).

Line 22: apropo my earlier comment you could change this to .. "...lowering the fear of humans in other species, be they kept as livestock, laboratory or as pet animals"

This was modified as suggested (line 23 to 24).

Line 41: please address this query in your methods rationale, and in discussion. Why was there no baseline data taken of docility and fear tests prior to the presumed (and evidenced) stressful experience of harvesting and clipping.

We have clarified why we started with the physiological stress response tests followed by the behavioural tests (line 146 to 149; line 189 to 191; line 219 to 220 and; line 251 to 253). Behavioural tests at an earlier age (7.5 months or before) may not have been accurate enough to predict the behaviour in ostriches when they reach the puberty stage. This is because handling difficulties and aggressive behaviour in ostriches starts to be exhibited later at puberty.

Line 45: again... to be addressed in method / discussion. Who did the harvesting / clipping... the familiar handler or an unfamiliar handler.

We have clarified as suggested throughout the methods section (i.e. line 148 to 149; line 192 to 193; line 262 to 263)

AND in all the (repeated) reactivity tests.. please make clear if the Unfamiliar handler actually was unfamiliar....  they would not be unfamiliar if had been involved in harvesting /clipping, routine care, or had been used in any of the previous docility / reactivity tests. This is not clear here nor in the Applied Animal Behaviour Science paper cited.

We have clarified as suggested throughout the methods section (i.e. line 148 to 149; line 151 to 153; line 192 to 193; line 262 to 263)

Line 101: what does regular mean? what does extensive mean? more detail needed, or if published elsewhere , then please cite.

We have clarified as suggested (line 130 to 136).

Line 102: where on body? head?

We have clarified as suggested (line 112).

Line 109: this is not clear to me. Are you saying week two they get visited 6-7 am 8-9am etc  Week 3 6-7am then 9-10 am 12-1 pm etc?

We have clarified this as suggested (line 130 to 136).

Line 112: This section on how to prepare samples once taken is very detailed. However, how the samples were 'developed' i.e. the above 2.2. Treatments section is not sufficiently detailed to allow replication.

We have clarified as suggested (line 109 to 141).

More detail is needed, consider re-write, tables / timelines, consider supplementary documents, even supplementary video of handling procedure...

We have added the time table used over the weeks of treatment as an appendix for clarification.

Line 152: I make this some 90 repetitions of this test. Did you have 90 unfamiliar handlers? if not, then they were not unfamiliar.

ditto for the docility and the fear tests.

If these were not true unfamiliar people, how was this controlled for in the analysis.

This just needs more detail for clarification

We have clarified as suggested (Line 192 to 193). Unfamiliar refers to the handler not involved in the treatments phase.

Line 155: so did this test take place in home environment? where--- near feeding area?  Same birds identified for each test?

We have clarified this as suggested (Line 194 to 198).

Line 352: you have only described one.. line 116 if you are referring to harvesting and clipping then say so.

We have clarified as suggested (Line 434.

Line 374: more than 'familiarization.. which suggest a neutral association. I suggest this is more of a positive classical association, where CS is person plus food trough = food (ucs);    and may be influenced by where the reactivity test was conducted in the home pen. Same in docility and fear tests; as opposed to actually testing reaction to unfamiliar person... esp fear test where food trough present. Perhaps additional data to have collected that may have been more indicative could have been e.g. behaviour / duration of transfer from holding pen to test pen by familiar or unfamiliar person in both of the reactivity tests (docility and fear tests)

We have reworded this argument as suggested (Line 436 to 444).

Line 393: yes.. and perhaps not unexpected in a species that uses vision to discriminate between fine detail of edible / non edible food items  (cattle may be more scent orientated)  :)  It would be nice for you to consider this in terms of the species' ethology...

We have added this argument as suggested (Line 477 to 480).

Line 402: are required? are you doing them?

We have reworded this sentence as we could not performed such tests as the birds were either transferred to breeders camps with other birds for production purposes or slaughtered (Line 488 to 490)

Line 411: anin fear test food was present.. these birds have had multitudinous opportunities to form a classical association between 'person as CS and food as UCS... so the test may simply be testing  this classical learning and not 'fear' as such.

d a further point for you to include ...

We have added this argument as suggested (line 506 to 509).

Line 430: do you mean constant? or do you mean intensive early gentle handling followed by consistent and frequent positive interactions helps develop better human-animal relationship and enhances the bird's ability to cope with stressful events such as feather harvesting.

We have reworded this sentence as suggested (line 509-511).

Round 2

Reviewer 2 Report

Ostrich Revised Version.

Thank you for this, it is much improved.  A few minor points that need a little more addressing.

Line 41-45  for clarity and conciseness change to   “Birds’ behavioural response towards familiar and unfamiliar human  was evaluated at 12 months using docility and fear tests, and by behaviour observations conducted three times per week between the ages of 8-13 months, which recorded willingness to approach, to allow touch interactions and exhibition of sexual display towards the person. “

Line 63 “occupational” is redundant

Line 88 replace “as well as” with “and”

Line 96:  please clarify (N=150 pairs)  as all other N in the document refer to individual birds.

Line 124 change “avoid potential discrimination” to “reduce potential discrimination”  (unlikely to be sufficient to avoid it!!)

Line 156 change “latter author” to “last author” … latter in this sentence actually would refer to the authors of citation 21

Line 207 “randomly” should be “random”

Line 205-208 This still needs a bit more clarity. The way it reads is that 2 individuals did all the testing.

each handler did three tests on a test day (morning afternoon and evening) i.e. an hour of observations for each.

This was done 3 times a week over 6 months ( months 8-13 inclusive)

By my maths that is  3x3x4 test sessions per month for each handler, i.e. 216 test sessions each

So where do the figures 51 and 55 come from? It does not even add up to enough sessions!  (ps even if only 1 a day, 3 times a week, and accounting for some 5 week months.. it is unlikely to be 106 sessions J)

Do you mean Line 207 that either the familiar handler OR the unfamiliar handler was randomly allocated to each test session. Three sessions were conducted each day (morning, afternoon and late afternoon).

As an aside what was the minimum interval between sessions?

OR do you mean Line 207

One session was conducted per day, and it was randomly allocated as to whether it was with the familiar or the unfamiliar handler. On each of the three test days per week the session was either in the morning, afternoon or late afternoon. 51 sessions were performed by familiar handler and 55 by the unfamiliar handler.

Line 220  if random then some birds may not have done the test and some done it twice?

Line 224 .. if identified at this point as having already been tested, was their data ignored?

Line 232.. surely this should be the familiar and the unfamiliar handler… you say it was the same 2 individual people as per the observation test.

Line 234 -236 for clarity I suggest reword sentence “The time taken….evaluate docility” to

“Latency to enter the marked square and duration of remaining there were recorded. The rationale for this test…..”

Line 249 – 255 suggest reword as

The fear test was adapted from that described by Mazurek et al. [26]. It was performed only on birds from 2015 batch who reached 12 months (N=…………….). As with the docility test, this was timed to correspond to the period when the birds reach puberty [27], and was conducted a week after the docility test ended. All birds underwent the fear test twice, once with the same familiar handler and once with the same unfamiliar handler. The surface of the test pen……..

Please note here…the word same is in italics: you need to clarify :  are these the same two individuals as per reactivity and docility test? If so, say so. If not, say so.

Line 282: so the binary format did not differentiate in the analysis if the interaction was aggressive or not? Really…. Can you clarify?

Line 282 :  You need to point out that for ALL tests the SAME TWO people were present.. either doing the test or recording it!  You need to say this in the discussion. . be more upfront about the methodological restrictions of this study.

Line 431: you should speculate why harvesting was more stressful. Likely has more pain associated with it.

Line 463. Does latter here refer to Bonato et al ? (you tend to overuse the word ‘latter’ it is not always clear”.  Were Bonato ‘foster parents’ birds?

I suggest you rewrite the sentence on lines 461 (It is notable…) to 466 (…presence.) as follow.

“This is supported by the findings of Bonato et al. [16] who found the largest difference between treatment groups in their study were between chicks reared by foster parent birds and pen reared chicks exposed to minimal human interaction during basic husbandry practices of supplying food and water.”

Line 469  please replace “inclination to approach the former handler rather than the latter” with “inclination to approach the familiar handler”.

Line 480 please change  “spend most of their time foraging…..” to “spend most of their time selectively foraging….”

To save repetition and aid clarity.. please change lines 495-500 as follows.

“…..temperament of birds at 12 months of age. This lack of effect on ostrich docility and fear may be explained in several ways. First, the effect of early human interactions may not persist over the post treatment – pre-test period from 3 – 12 months. Second, any effect may have been over-ridden by the stress of the procedure immediately prior to the test, namely being caught by an unfamiliar catching crew. Thirds, as the test used were adapted from………”

Line 512.. be really clear here, you need to be explicit about the methodological drawbacks of you study.! I suggest this sentence needs expanding to three sentences.

“Future research should consider two methodological aspects of the current study in particular. First is in refining the docility and fear test to make them more suited to ostrich. Second is to be more rigorous in the exposure birds have to humans. This is difficult on a commercial set-up, however in the current study, the birds were repeatedly tested by the same ‘unfamilar’ person, which compromises the definition of ‘unfamilar’ and potentially also the data.

Line 538 – 542. These last 2 sentences need transposing. I.e.

….ostrich temperament later in life. It might be that the frequency of human-animal interactions during the repeated behavioural (reactivity) testing both over-shadowed the early treatment effects and thus influenced the results of the later docility and fear tests.  That notwithstanding, the data indicates a need for further research in to the effects of early treatment on the activity of the HPA axis at 12 months, and the need to refine behaviour testing for this species. This will facilitate improved welfare, and potentially improved selection for breeding stock.

Author Response

Ostrich Revised Version.

Thank you for this, it is much improved.  A few minor points that need a little more addressing.

 Line 41-45  for clarity and conciseness change to   “Birds’ behavioural response towards familiar and unfamiliar human  was evaluated at 12 months using docility and fear tests, and by behaviour observations conducted three times per week between the ages of 8-13 months, which recorded willingness to approach, to allow touch interactions and exhibition of sexual display towards the person. “

This was changed in the text as suggested (line 41 to 44).

Line 63 “occupational” is redundant

The word was removed as suggested (line 63).

Line 88 replace “as well as” with “and”

This was modified as suggested (line 88).

Line 96:  please clarify (N=150 pairs)  as all other N in the document refer to individual birds.

This was clarified as suggested (line 96).

Line 124 change “avoid potential discrimination” to “reduce potential discrimination”  (unlikely to be sufficient to avoid it!!)

This was changed as suggested (line 124).

Line 156 change “latter author” to “last author” … latter in this sentence actually would refer to the authors of citation 21

This was changed as suggested. The sentence has been rephrased (line 155 to 157).

Line 207 “randomly” should be “random”

This was changed as suggested (line 207).

Line 205-208 This still needs a bit more clarity. The way it reads is that 2 individuals did all the testing. each handler did three tests on a test day (morning afternoon and evening) i.e. an hour of observations for each.This was done 3 times a week over 6 months ( months 8-13 inclusive)

By my maths that is  3x3x4 test sessions per month for each handler, i.e. 216 test sessions each

So where do the figures 51 and 55 come from? It does not even add up to enough sessions!  (ps even if only 1 a day, 3 times a week, and accounting for some 5 week months.. it is unlikely to be 106 sessions J)

The observations were conducted on 106 days chosen at random when birds were between the age of 8 to 13 months of age. This was clarified in the text (line 189 to 190 and line 208 to 210).

Do you mean Line 207 that either the familiar handler OR the unfamiliar handler was randomly allocated to each test session. Three sessions were conducted each day (morning, afternoon and late afternoon).As an aside what was the minimum interval between sessions?OR do you mean Line 207One session was conducted per day, and it was randomly allocated as to whether it was with the familiar or the unfamiliar handler. On each of the three test days per week the session was either in the morning, afternoon or late afternoon. 51 sessions were performed by familiar handler and 55 by the unfamiliar handler.

This was clarified in the text (line 208 to 210). Yes, either the familiar OR the unfamiliar handler was allocated to each test session.  Only one session was conducted per day, either in the morning, afternoon or late afternoon; on total of 51 sessions performed by a familiar handler and 55 by the unfamiliar handler. 

Line 220  if random then some birds may not have done the test and some done it twice?

This was clarified in the text. The word random has been removed and we explained that all birds were tested once by a familiar and unfamiliar handler. Also we explained how tested birds were moved to a different pen to avoid re testing them (line 227 to 230).

Line 224 .. if identified at this point as having already been tested, was their data ignored?

This was clarified in the text. If the bird have been already tested, the bird was released without been tested again as they were identified before the test (line 227 to 230).

Line 232.. surely this should be the familiar and the unfamiliar handler… you say it was the same 2 individual people as per the observation test.

Yes, the same individual people who performed the observations performed all other tests. This was clarified in the text (docility test: line 232 to 233; fear test: line 258 to 260).

Line 234 -236 for clarity I suggest reword sentence “The time taken….evaluate docility” to“Latency to enter the marked square and duration of remaining there were recorded. The rationale for this test…..”

This was modified as suggested (line 241).

Line 249 – 255 suggest reword as

The fear test was adapted from that described by Mazurek et al. [26]. It was performed only on birds from 2015 batch who reached 12 months (N=…………….). As with the docility test, this was timed to correspond to the period when the birds reach puberty [27], and was conducted a week after the docility test ended. All birds underwent the fear test twice, once with the same familiar handler and once with the same unfamiliar handler. The surface of the test pen……..

This was modified as suggested (line 254 to 259).

Please note here…the word same is in italics: you need to clarify :  are these the same two individuals as per reactivity and docility test? If so, say so. If not, say so.

This was clarified as requested (line 258 to 259).

 Line 282: so the binary format did not differentiate in the analysis if the interaction was aggressive or not? Really…. Can you clarify?

This was clarified as suggested (line 286; line 428 to 429 and 443 to 445).

Line 282 :  You need to point out that for ALL tests the SAME TWO people were present.. either doing the test or recording it!  You need to say this in the discussion. . be more upfront about the methodological restrictions of this study.

 This was clarified for all the tests as suggested (line 232 to 233; line 238 to 239; line 259 to 260).

Line 431: you should speculate why harvesting was more stressful. Likely has more pain associated with it.

 This was added in the text (line 464 to 467).

Line 463. Does latter here refer to Bonato et al ? (you tend to overuse the word ‘latter’ it is not always clear”.  Were Bonato ‘foster parents’ birds?

This was clarified as suggested. The word latter has been removed and the statement rephrased (line 501).

I suggest you rewrite the sentence on lines 461 (It is notable…) to 466 (…presence.) as follow.

“This is supported by the findings of Bonato et al. [16] who found the largest difference between treatment groups in their study were between chicks reared by foster parent birds and pen reared chicks exposed to minimal human interaction during basic husbandry practices of supplying food and water.”

This was modified for clarification on the results of the present study compared to those of Bonato et al. [16] (line 500). Please note that in Bonato et al. [16] the largest difference between treatment groups were between chicks reared by foster parent birds and pen reared chicks exposed to extensive human presence.

Line 469  please replace “inclination to approach the former handler rather than the latter” with “inclination to approach the familiar handler”.

This was replaced as suggested (line 506).

Line 480 please change  “spend most of their time foraging…..” to “spend most of their time selectively foraging….”

 This was changed as requested (line 517).

To save repetition and aid clarity.. please change lines 495-500 as follows.

“…..temperament of birds at 12 months of age. This lack of effect on ostrich docility and fear may be explained in several ways. First, the effect of early human interactions may not persist over the post treatment – pre-test period from 3 – 12 months. Second, any effect may have been over-ridden by the stress of the procedure immediately prior to the test, namely being caught by an unfamiliar catching crew. Thirds, as the test used were adapted from………”

 This was modified as suggested (line 533 to 537).

Line 512.. be really clear here, you need to be explicit about the methodological drawbacks of you study.! I suggest this sentence needs expanding to three sentences.

“Future research should consider two methodological aspects of the current study in particular. First is in refining the docility and fear test to make them more suited to ostrich. Second is to be more rigorous in the exposure birds have to humans. This is difficult on a commercial set-up, however in the current study, the birds were repeatedly tested by the same ‘unfamilar’ person, which compromises the definition of ‘unfamilar’ and potentially also the data.

This was modified for clarification on the drawbacks of the study (line 548 to 554).

Line 538 – 542. These last 2 sentences need transposing. I.e.

….ostrich temperament later in life. It might be that the frequency of human-animal interactions during the repeated behavioural (reactivity) testing both over-shadowed the early treatment effects and thus influenced the results of the later docility and fear tests.  That notwithstanding, the data indicates a need for further research in to the effects of early treatment on the activity of the HPA axis at 12 months, and the need to refine behaviour testing for this species. This will facilitate improved welfare, and potentially improved selection for breeding stock.

The sentences were transposed in the text as suggested (line 579 to 585).